

# Utility of a time frame in assessing psychological pain and suicide ideation

Esther L. Meerwijk[1,3] and Sandra J. Weiss[2]

[1] Department of Family Health Care Nursing, University of California, San Francisco, CA, United States of America

[2] Department of Community Health Systems, University of California, San Francisco, CA, United States of America

[3] Current affiliation: VA Health Services Research & Development (Center for Innovation to Implementation), VA Palo Alto Health Care System, Menlo Park, CA, United States of America

## ABSTRACT

**Background.** Assessing psychological pain has been recommended as an integral part of a comprehensive suicide risk assessment. The Psychache Scale, an established measure of psychological pain, does not specify a time frame for when pain is experienced, which may inadvertently increase the likelihood of identifying individuals as being at elevated suicide risk when they complete the Psychache Scale based on psychological pain experienced at some undefined time in the past.

**Methods.** We conducted a national general population survey among United States adults to determine whether addition of a time frame to the instructions of the Psychache Scale would more accurately reflect current psychological pain and more effectively identify people with current suicide ideation. A between-subjects design was used where respondents were randomized to complete the original Psychache scale or a modified scale with time frame. Data were collected online from September 2015 to June 2016. A total of 242 respondents had complete psychological pain data: 133 completed the original Psychache Scale and 109 completed the Psychache Scale with time frame.

**Results.** Addition of a time frame did not result in differences in psychological pain scores. However, when screening for participants with current suicide ideation, 13% fewer false positives were observed with the modified scale at higher cut-off values than previously reported (38 vs. 24). The substantial increase in positive predictive value suggests that a time frame is a worthwhile addition to the Psychache Scale.

**Discussion.** We recommend using the Psychache Scale with a time frame and testing the cut-off score for suicide ideation in population samples that reflect the general population more accurately. Psychological pain cut-off scores in clinical samples have yet to be established.

Corresponding author
Esther L. Meerwijk,
esther.meerwijk@gmail.com,
esther.meerwijk@va.gov

## INTRODUCTION

Theories of suicide often mention escape from an intolerable or unbearable state of mind as the objective of suicidal behavior (*Baumeister, 1990*; *Shneidman, 1999*; *Maltsberger, 2004*). Many if not most people who find themselves in this state experience psychological pain

with which they can no longer cope. Psychological pain has been defined as "a lasting, unsustainable, and unpleasant feeling resulting from negative appraisal of an inability or deficiency of the self" (*Meerwijk & Weiss, 2011*) and is a significant risk factor for suicide in people with and without a mental health diagnosis (*Verrocchio et al., 2016*; *Rizvi et al., 2017*). As predicting suicide attempts remains notoriously difficult (*Turecki & Brent, 2016*), there is a need to explore new avenues to enhance the validity of suicide risk assessments. One method to accomplish this may be the assessment of psychological pain (*Verrocchio et al., 2016*).

The Psychache Scale (*Holden et al., 2001*) is among the most frequently used instruments to assess psychological pain today. The PS has been used in clinical as well as general population samples and was found to distinguish participants with respect to their level of suicide ideation and history of suicide attempts. In a direct comparison of the PS with the Beck Depression Inventory (BDI) and the Beck Hopelessness Scale (BHS), the PS significantly outperformed the BDI and BHS in terms of identifying individuals with a history of multiple suicide attempts including an attempt in the past year (*Troister, D'Agata & Holden, 2015*). Studies in undergraduate students, homeless men, and prison inmates found that psychological pain was significantly associated with a history of suicide attempts (*Pereira et al., 2010*; *Troister & Holden, 2010*; *Patterson & Holden, 2012*; *You et al., 2014*). In a nonclinical convenience sample of adults, psychological pain was directly related to suicide risk, as assessed on the Suicidal Behavior Questionnaire—Revised (*Campos & Holden, 2015*), which combines self-reported past suicidal thoughts and behavior, current suicidal thoughts, and the likelihood of a future suicide attempt in a single score. Medium to strong positive correlations were found between psychological pain and suicide ideation in undergraduate students, healthy people, and patients with depression (*Holden et al., 2001*; *Li et al., 2014*; *Xie et al., 2014*). Psychological pain was also significantly higher in depressive patients with suicide ideation than in patients without suicide ideation (*Cáceda et al., 2014*). Moreover, rather than depression or hopelessness, it was the change in psychological pain after two years that predicted increased suicide ideation in undergraduate students who experienced suicide ideation at baseline and had a history of attempted suicide at baseline (*Troister & Holden, 2012*).

While these studies provide empirical support for the relationship between psychological pain and suicidal thoughts and behaviors, the PS score classified a high number of undergraduate students at elevated risk for suicide who were not at risk based on their current suicidal thoughts and suicide attempt history (*Troister, D'Agata & Holden, 2015*). In a related area of research, biological markers in adults with a history of depression were not associated with psychological pain assessed on the PS, whereas these markers were associated with an alternate measure of psychological pain, namely the Orbach & Mikulincer Mental Pain (OMMP) questionnaire (*Meerwijk, Chesla & Weiss, 2014*; *Meerwijk, Ford & Weiss, 2015*). An important difference between the OMMP and PS is that the OMMP assesses current psychological pain, whereas the PS allows respondents to reflect on their lifetime psychological pain. This difference may not be apparent with patients who are experiencing significant psychological pain when they complete these measures, because they are likely to focus on their current feelings. However, when patients

who do not currently experience significant psychological pain complete the PS and reflect on psychological pain they experienced at some undefined time in the past they may report higher psychological pain and be misclassified as being at elevated risk for suicide. It has been suggested that the PS's utility in suicide risk assessments could be enhanced by adding a time frame to the scale's instructions (*Meerwijk & Weiss, 2016*).

The study presented here had two primary aims. The first aim was to determine if adding a time frame to an otherwise unaltered PS, hereon referred to as PS-TF, would affect psychological pain scores when compared to the original PS. We used a between-subjects design where study participants were randomized to complete either the original PS or the PS-TF. We hypothesized that for participants who reported psychological pain during the past week, there would be no significant difference between scores on the PS-TF and the PS, whereas PS-TF scores would be lower than PS scores when psychological pain was reported longer than a week ago. The second aim was to determine if the PS-TF was more effective in identifying individuals with suicide ideation than the PS. We expected a lower number of false positives with the PS-TF and hypothesized higher specificity and positive predictive power with the PS-TF compared to the PS. Addressing these hypotheses will further our understanding of psychological pain and its clinical utility in suicide risk assessments.

## METHODS

### Participants

We collected data from general population adults who responded to Facebook advertisements that invited to complete an anonymous online survey. A similar approach was used to collect data on suicidal behavior in an Australian online sample (*Batterham, Calear & Van Spijker, 2015*). To draw people's attention, ad texts focused on the link between psychological pain and suicide, the prevalence of suicide and suicide attempts, or that someone close may be experiencing unbearable pain. Another ad indicated that people of any race/ethnicity could participate, regardless of whether they were experiencing psychological pain. In addition, all ads showed a map of the United States (US) and indication that this was a national survey. Inclusion criteria were being at least 18 years old, a resident of the US, and able to read and understand English. Of 708 individuals who visited the survey website, 389 provided consent (55%) to participate. Among those who consented, 133 individuals did not submit their responses (34%). Final analysis was done on 242 respondents with complete psychological pain data: 133 completed the PS and 109 completed the PS-TF.

A comparison of respondents who submitted their data and respondents who dropped out before completing the survey showed no statistical differences in terms of gender, ethnicity, sexual orientation, student status, when they had last experienced psychological pain, when they had experienced their worst psychological pain, whether they wished to be dead during the past week, or whether they had ever attempted suicide. However, respondents who submitted their data were older (32.5 [*SD* 15.3] vs. 29.7 [*SD* 14.4], $p = .082$) and significantly more likely to have used pain medication on the day they completed the survey ($\chi^2 = 5.10, p = .024$), to have been diagnosed with a mental illness ($\chi^2 = 4.51, p = .034$), and to have lost someone to suicide ($\chi^2 = 7.95, p = .005$).

## Measures

The survey started with demographic questions and then asked about medication use, psychological pain, tolerance for psychological pain, suicide history, and feelings about death and dying, in order of what was considered increasing sensitivity (i.e., content that participants could perceive as potentially threatening or problematic). Psychological pain tolerance and feelings about death and dying are central to a forthcoming article and will not be discussed here.

### Psychache scale

To orient respondents, the PS describes psychological pain as "a hurting feeling inside, often described as pain you feel in your heart or mind. It indicates how much you hurt emotionally or mentally" (*Holden et al., 2001*). The PS instructions then continued with "The following statements refer to your psychological pain, NOT your physical pain. Please indicate how frequently each of the following occurs". The PS has 13 items with higher scores reflecting greater psychological pain. Nine items are scored on a 5-point frequency scale ranging from never to always, corresponding to a value of 1–5. Four more items, reflecting pain intensity, are scored on a 5-point symmetrical scale ranging from strongly disagree to strongly agree, also corresponding to a value of 1–5. The total score is obtained by summing the item scores, resulting in a total score between 13 and 65. The original paper-version of the PS is shown in the 'Appendix'.

The PS is well validated in diverse populations, including university students, homeless men, outpatients with depression, and male prison inmates (*Mills, Green & Reddon, 2005*; *Patterson & Holden, 2012*; *Troister & Holden, 2012*; *Li et al., 2014*; *Xie et al., 2014*). We found excellent internal consistency reliability for our sample with Cronbach's $\alpha = .96$ and no inter-item correlations lower than 0.43.

### Psychache scale with time frame

The PS-TF uses the same items as the original PS; however, to have respondents focus on current psychological pain, we modified the last sentence of the instructions at the top of the scale to read "Please indicate how frequently each of the following *has been occurring during the past week, including today*". This time frame was chosen because psychological pain is a lasting state that takes time for resolution (*Meerwijk & Weiss, 2011*) and we assumed psychological pain to linger with varying intensity for days at least. Moreover, the widely used Beck Scale for Suicide ideation (*Beck, Brown & Steer, 1997*) uses the same time frame to assess thoughts of suicide and suicidal behavior. We found excellent internal consistency reliability for the PS-TF with Cronbach's $\alpha = .94$ and no inter-item correlation lower than 0.37.

### Suicidal ideation and behavior

We used respondents' wish to be dead during the past week as a dichotomous measure of suicide ideation (yes/no). The 'wish to be dead' criterion often features among initial questions to assess suicide risk; for example, in the Columbia Suicide Severity Rating scale (*Posner et al., 2011*) and in the MINI Neuropsychiatric interview (*Sheehan et al., 1998*). In addition, we asked whether they had attempted suicide and if so, how often and how long ago for the most recent attempt (past week, month, year, or longer ago).

## Procedure

Data were collected from September 2015–June 2016. We used Research Electronic Data Capture (REDCap$^{TM}$) to develop our online survey and collect the data. REDCap$^{TM}$ is a secure and HIPAA-compliant electronic data capture tool (*Harris et al., 2009*). Visitors to the survey web site first arrived on the informed consent page, describing the nature of the survey and resources available to them should the survey make them feel uncomfortable or upset (e.g., the National Suicide Prevention Lifeline phone number). By submitting their age, city, and ZIP code on the consent page, respondents self-identified as being eligible and initiated the actual survey. Based on the time of written consent in seconds (odd/even), participants were assigned to a group that completed the original PS or a group that completed the PS-TF.

Upon completion of the survey, respondents were offered a choice to participate in a drawing to win a $50 debit card. As this was an anonymous survey and respondents could submit their data more than once, even though they were instructed not to, we developed an algorithm to search the data for potential duplicate entries. The algorithm compared age, gender, city, and ZIP code. We then manually compared these entries and retained the first entry without missing data. Of potential duplicates identified by the algorithm, five entries were considered actual duplicates and discarded. The Institutional Review Board at the University of California, San Francisco, approved the study (IRB# 16-18686).

## Data analysis

We calculated means (*SD*) and frequencies of sociodemographic variables for participants who completed the PS or PS-TF and tested for differences between the two groups using Student's *t* and Pearson's $\chi^2$. Missing data were omitted pairwise. Depending on when psychological pain was last experienced, pain scores were right skewed, violating assumptions for a parametric model. Transforming the data did not satisfactorily normalize their distribution. Therefore, we used Wilcoxon rank sum tests (also known as Mann–Whitney) to test our first hypothesis regarding differences in PS and PS-TF scores when participants reported pain experienced currently versus during more distant time frames. Scores were examined for psychological pain experienced "today", "during the past week", and "more than a week ago". We applied a Bonferroni correction to reduce the risk of type-I error related to multiple testing.

Hypothesis two was assessed by comparing the scales' efficiency in distinguishing respondents with and without suicide ideation. We determined the PS and PS-TF's sensitivity, specificity, and positive and negative predictive value (PPV, NPV) at multiple cutoff values. Sensitivity was calculated as True Positives (TP)/(TP + False Negatives (FN)) and specificity was calculated as True Negatives (TN)/(TN + False Positives (FP)). Positive predictive value was calculated as TP/(TP + FP) and negative predictive value was calculated as TN/(TN + FN). The 95% confidence intervals (CI) for the difference score between the two scales were determined with a bootstrap approach for each of these metrics (*Altman, 2000*). We report results obtained with 10,000 steps and the basic bootstrap method to calculate CIs, but similar values were obtained with other methods. Screening efficiency is visualized in a

receiver-operating curve (ROC) and optimum cutoff values were determined by minimizing the Euclidian distance between the top left of the ROC plot and the respective curves. R version 3.2.4 was used for all analyses, and statistical significance was assumed at $p < .05$.

## RESULTS

### Sample description

Table 1 shows sociodemographic characteristics of respondents. No statistically significant differences were observed between the PS group and the PS-TF group, although the racial background of respondents randomized to the PS group was more often white (84 vs. 58, $\chi^2 = 2.736, p = .098$). Overall, respondents were more often female (59%) than male (39%). We asked respondents questions to determine whether they belonged to specific "vulnerable populations". Among them, few respondents indicated being pregnant ($n = 4$), being imprisoned ($n = 1$), or being hospitalized ($n = 2$) at the time they completed the survey. With respect to sexual orientation, 47 respondents (20%) described themselves as gay/lesbian, bisexual, asexual, or unsure about their sexual orientation. With respect to mental illness, 50 respondents (21%) indicated having been diagnosed with a mental illness. Sixty-one respondents (25%) indicated having attempted suicide at least once, seventeen of which (7%) made an attempt during the past year. Forty respondents (17%) indicated having lost someone close to suicide. With respect to pain medication, 62 respondents (26%) indicated being on pain medication on the day they completed the survey.

Compared to the 2010 US census records (US Census Bureau, 2016), our sample was younger and more often female. Regarding race, our sample was less often of white or Asian descent and more often of Black or African American, American Indian/Alaska Native, Native Hawaiian and other Pacific Islander, or of mixed descent. Among US adults, 0.6% reported having attempted suicide during the past year and 18.5% reported having any mental illness (Substance Abuse and Mental Health Services Administration, 2014), against 7% and 21% of respondents in our survey, respectively.

### Hypothesis testing for original PS versus PS-TF

The majority of participants reported psychological pain during the past week, including today (see Table 1). The overall mean PS and PS-TF scores were 30.2 (SD 13.4) and 31.6 (SD 12.7), respectively. Hypothesis 1 was not supported. A Wilcoxon rank-sum test among participants who experienced psychological pain longer than a week ago, showed no statistically significant difference in psychological pain ($W = 739, p = .150$) between the PS group ($Mdn = 16, n = 44$) and the PS-TF group ($Mdn = 19, n = 41$). Cliff's delta, an ordinal measure of effect size (Cliff, 1993), suggested a small effect (delta = 0.181). As expected among participants who experienced psychological pain within the past week, we found no statistically significant differences in psychological pain between the PS group and the PS-TF group (see Fig. 1).

Overall, 56 respondents (23%) indicated a wish to be dead during the past week. Table 2 shows their psychological pain scores. The scales' screening efficiency in terms of identifying respondents with and without a wish to be dead is visualized in Fig. 2. Screening accuracy, as indicated by the area under the ROC (AUC), was similar for both scales (.785 and

**Table 1  Characteristics for groups who completed psychological pain scales with different time frame instructions.**

| | PS ($n = 133$) | PS-TF ($n = 109$) | Total ($N = 242$)[a] | |
|---|---|---|---|---|
| Mean age in years (*SD*) | 33.1 (15.6) | 31.9 (15.3) | 32.6 (15.4) | $t = 0.594$ |
| Gender | | | | $\chi^2 = 4.506$[b] |
|    Male | 52 (39.1%) | 43 (39.8%) | 95 | |
|    Female | 77 (57.9%) | 64 (59.3%) | 141 | |
|    Transgender | 0 (0%) | 1 (0.9%) | 1 | |
|    Do not use either label | 4 (3.0%) | 0 (0%) | 4 | |
| Race | | | | $\chi^2 = 4.760$[b] |
|    American Indian/Alaska native | 5 (4.3%) | 4 (4.2%) | 9 | |
|    Asian | 1 (0.9%) | 4 (4.2%) | 5 | |
|    Native Hawaiian/other PI | 1 (0.9%) | 1 (1.1%) | 2 | |
|    Black/African American | 15 (12.8%) | 18 (18.9%) | 33 | |
|    White | 84 (71.8%) | 58 (61.1%) | 142 | |
|    Mixed | 11 (9.4%) | 9 (10.5%) | 20 | |
| Hispanic/Latino ethnicity | | | | $\chi^2 = 0.041$ |
|    No | 95 (72.0%) | 79 (73.1%) | 174 | |
|    Yes | 37 (28.0%) | 29 (26.9%) | 66 | |
| English as first language | | | | $\chi^2 = 1.115$ |
|    No | 18 (13.5%) | 20 (18.5%) | 38 | |
|    Yes | 115 (86.5%) | 88 (81.5%) | 203 | |
| College/University student | | | | $\chi^2 = 0.197$ |
|    No | 95 (71.4%) | 75 (68.8%) | 163 | |
|    Yes | 38 (28.6%) | 34 (31.2%) | 72 | |
| Experienced psychological pain | | | | $\chi^2 = 0.472$ |
|    Today | 44 (33.1%) | 39 (35.8%) | 83 | |
|    During past week | 45 (33.8%) | 29 (26.6%) | 74 | |
|    Longer ago | 44 (33.1%) | 41 (37.6%) | 85 | |

Notes.
  PS, Psychache scale; PS-TF, Psychache scale with time frame; PI, Pacific Islander.
  [a]Counts do not always add up to *N* due to missing data.
  [b]Test included cells with expected counts less than 5.

**Table 2  Psychological pain scores by Psychache Scale version and current wish to be dead.**

| | Wish to be dead | | | | |
|---|---|---|---|---|---|
| | **Yes** | | | **No** | |
| | *n* | Mean (*SD*) | | *n* | Mean (*SD*) |
| PS | 27 | 42.1 (14.3) | | 106 | 27.1 (11.3) |
| PS-TF[a] | 29 | 40.7 (11.1) | | 79 | 28.2 (11.6) |
| Overall | 56 | 41.4 (12.7) | | 185 | 27.6 (11.4) |

Notes.
  PS, Psychache Scale; PS-TF, Psychache Scale with Time Frame.
  [a]Missing wish to be dead for one participant.

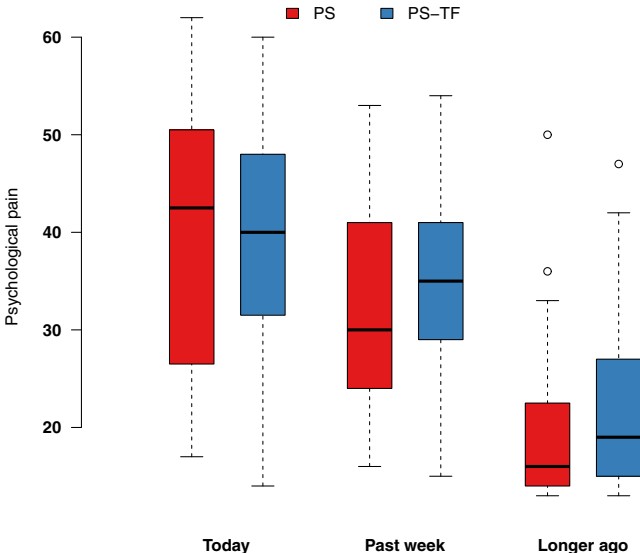

**Figure 1** **Box plot of psychological pain scores differentiated by PS and PS-TF and when pain was last experienced.** Whiskers represent 1.5 times the interquartile range. Circles indicate outliers. PS, Psychache Scale; PS-TF, Psychache Scale with time frame.

**Table 3** Screening efficiency at multiple cut-off values for a current wish to be dead.

|  | Sensitivity | Specificity | PPV | NPV |
|---|---|---|---|---|
| **PS** |  |  |  |  |
| ≥35 | .78 | .75 | .45 | .93 |
| ≥**36** | .78 | .75 | .45 | .93 |
| ≥37 | .74 | .76 | .44 | .92 |
| ≥38 | .70 | .78 | .45 | .91 |
| ≥39 | .67 | .78 | .44 | .90 |
| **PS-TF** |  |  |  |  |
| ≥35 | .76 | .73 | .51 | .89 |
| ≥36 | .72 | .80 | .57 | .89 |
| ≥37 | .72 | .80 | .57 | .89 |
| ≥**38** | .72 | .81 | .58 | .89 |
| ≥39 | .69 | .82 | .59 | .88 |

**Notes.**
Bold face cut-off values indicate optimum combination of sensitivity and specificity.
PPV, positive predictive value; NPV, negative predictive value; PS, Psychache Scale; PS-TF, Psychache Scale with time frame.

.784 for the PS and PS-TF, respectively). Table 3 shows screening metrics for both scales against multiple cutoff values. The optimum cut-off value for a wish to be dead was 36 for the PS and 38 for the PS-TF. As can be seen from Table 3, specificity and PPV at the optimum cutoff value were higher for the PS-TF (.81 and .58) than for the PS (.75 and .45), albeit at a 6% reduction in sensitivity and a 4% reduction in negative predictive value (.93 vs. .89). 95% CIs for difference scores did not indicate statistically significant differences (sensitivity: −.19–.29, specificity: −.18–.07, positive predictive value: −.35–.09, negative predictive value: −.05–.14).

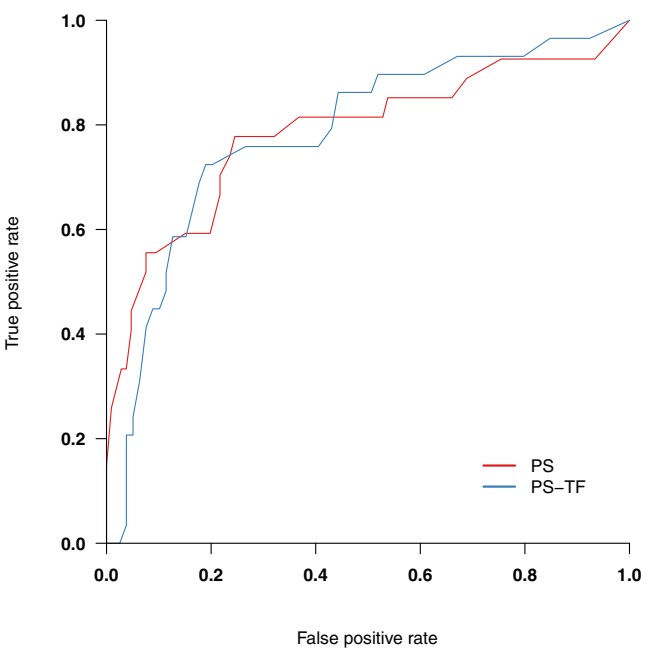

**Figure 2** **Receiver-operating curves for PS and PS-TF, screening for a current wish to be dead.** PS, Psychache Scale; PS-TF, Psychache Scale with time frame.

## DISCUSSION

We evaluated the effect of adding a time frame to the instructions of the PS on (1) psychological pain scores and (2) screening efficiency for suicide ideation. Our hypothesis that PS-TF scores would be lower than PS scores when psychological pain was reported longer than a week ago was not supported. The median PS-TF score was slightly higher, although the difference was not significant and represented a small effect size. The data did support our hypothesis that there would be no significant difference between scores on the PS and PS-TF for participants who reported psychological pain during the past week including today. This group represented the majority of participants, and it may be that the group who reported psychological pain longer than a week ago had limited variability. A within-subjects design in which participants would have completed both scales might have compensated for that. However, we specifically opted to have participants complete one psychological pain scale only, as completing one scale after the other could change participants' spontaneous response to whichever scale was completed last.

The data did support our hypothesis that specificity and positive predictive power when screening for suicide ideation, operationalized as wishing to be dead during the past week, would be higher for the PS-TF than for the PS. Although not statistically significant, measures differed 6% points in terms of specificity (.81 vs. .75) and 13% points in terms of PPV (.58 vs. .45) at the optimum cutoff values of 36 for the PS and 38 for the PS-TF. This implies 13% fewer false positives when screening for suicide ideation among general population adults with the PS-TF. In a comparable analysis of undergraduate students with suicide ideation, *Troister, D'Agata & Holden (2015)* reported a specificity of .79 and PPV

of .27 when using the PS. These values are respectively 2% points and 21% points lower than obtained with the PS-TF. Note that PPV depends on prevalence in the population and that the large difference in PPV can at least in part be explained by the higher prevalence of people with suicide ideation in our sample. While fewer false positives were observed with the PS-TF, it should be mentioned that this was accompanied by an increase in false negatives (4%). Further research is warranted to better understand any factors that are potentially being missed in this group of individuals at risk for suicide ideation. Still, the substantial increase in PPV acquired by adding a time frame to the measure suggests it is a worthwhile addition, especially since assessing psychological pain is only one aspect of a comprehensive suicide risk assessment. The difference in cutoff scores at which these metrics were observed, 24 in the Troister sample vs. 38 for the PS-TF, is also striking. In light of mean psychological pain scores in our sample and other nonclinical samples of adults (28.2 [*SD* 13.5] *Klonsky & May, 2015*; 24.5 [SD 11.6] *Campos & Holden, 2015*), the cutoff score of 24 that Troister et al. reported as a screen for suicide ideation seems low and prone to a high number of false positives in the general population. Our data indicate that cutoff scores of 36 for the PS and 38 for the PS-TF suggest clinically salient suicide ideation in general population adults.

With average psychological pain scores in the low thirties, our sample experienced a higher level of psychological pain than was observed in a large sample of undergraduate students (20.35 [SD 8.25] *Troister, D'Agata & Holden, 2015*). As expected, our observed scores were higher than those found previously in healthy adults (13.70 [SD 0.50] *Cáceda et al., 2014*) and (20.60 [SD 4.95] *Xie et al., 2014*), and not quite as high as the level of psychological pain observed in outpatients with major depressive episodes (40.01 [SD 11.86] *Li et al., 2014*). The nonclinical sample of adults referred to earlier (*Campos & Holden, 2015*) is perhaps the most comparable to our sample and our higher psychological pain scores may at least in part be explained by the relatively high percentage of people who had attempted suicide in the past year. Moreover, comparison of people who dropped out with people who completed the survey showed that people who completed the survey were more likely to have used pain medication on the day of completing the survey and more often had lost someone to suicide, further suggesting that our sample may have been more vulnerable than the US population in general. For these reasons and the observation that our sample was more often female, generalization of our results to the general US population should be made cautiously. In addition, our sample size did not allow statistical analysis of those who had a history of suicide attempt in addition to having a current wish to be dead. A history of suicide attempt is an important indicator of suicide risk (*Chu et al., 2015*). Yet, many commonly known risk factors have poor ability in distinguishing suicidal people who attempt suicide from people who are suicidal but never attempt suicide (*May & Klonsky, 2016*). While there is evidence that psychological pain distinguishes those with and without a history of suicide attempt (*Pereira et al., 2010*; *Troister & Holden, 2010*; *Patterson & Holden, 2012*; *You et al., 2014*), longitudinal research is needed to assess whether current psychological pain distinguishes those who will attempt suicide or die by suicide from those with suicide ideation who will never attempt suicide.

Our study was aimed at enhancing the applicability of the PS in suicide risk assessments by adding a timeframe to its instructions. While the effect was not immediately clear in pain scores themselves, the PS-TF more accurately identified participants with a current wish to be dead. We want to emphasize that we have no intention of predicting suicide risk based on a single measure. In concert with *Verrocchio et al. (2016)*, we recommend that psychological pain be assessed as one component in a more comprehensive suicide assessment. As more than two thirds of people with suicidal behaviors do not express their suicidal intent to healthcare professionals (*Turecki & Brent, 2016*), the PS/PS-TF provides an initial indication that suicide ideation may be present, without actually referring to suicide in its items (*Troister, D'Agata & Holden, 2015*). We also recommend that the cutoff score for psychological pain identified in our findings be tested in future research with general population samples that reflect the general population more accurately. Cutoff scores for psychological pain in clinical samples have yet to be established.

## APPENDIX

The Psychache Scale (*Holden et al., 2001*).

The following statements refer to your psychological pain, NOT your physical pain. By circling the appropriate number, please indicate how frequently each of the following occurs.

1 = Never; 2 = Sometimes; 3 = Often; 4 = Very Often; 5 = Always

| | | | | | |
|---|---|---|---|---|---|
| 1. I feel psychological pain. | 1 | 2 | 3 | 4 | 5 |
| 2. I seem to ache inside. | 1 | 2 | 3 | 4 | 5 |
| 3. My psychological pain seems worse than any physical pain. | 1 | 2 | 3 | 4 | 5 |
| 4. My pain makes me want to scream. | 1 | 2 | 3 | 4 | 5 |
| 5. My pain makes my life seem dark. | 1 | 2 | 3 | 4 | 5 |
| 6. I can't understand why I suffer. | 1 | 2 | 3 | 4 | 5 |
| 7. Psychologically, I feel terrible. | 1 | 2 | 3 | 4 | 5 |
| 8. I hurt because I feel empty. | 1 | 2 | 3 | 4 | 5 |
| 9. My soul aches. | 1 | 2 | 3 | 4 | 5 |

Please continue this inventory using the following scale:

1 = Strongly Disagree; 2 = Disagree; 3 = Unsure; 4 = Agree; 5 = Strongly Agree

| | | | | | |
|---|---|---|---|---|---|
| 10. I can't take my pain any more. | 1 | 2 | 3 | 4 | 5 |
| 11. Because of my pain, my situation is impossible. | 1 | 2 | 3 | 4 | 5 |
| 12. My pain is making me fall apart. | 1 | 2 | 3 | 4 | 5 |
| 13. My psychological pain affects everything I do. | 1 | 2 | 3 | 4 | 5 |

Note: for the Psychache Scale with time frame, we modified the second sentence of the instructions at the top of the scale to read "please indicate how frequently each of the following has been occurring during the past week, including today".

### Funding

This study was supported by the National Institute of Nursing Research grant T32 NR07088. There was no additional external funding received for this study. The funders had no role in study design, data collection and analysis, decision to publish, or preparation of the manuscript.

### Grant Disclosures

The following grant information was disclosed by the authors:
National Institute of Nursing Research: T32 NR07088.

### Competing Interests

The authors declare there are no competing interests.

### Author Contributions

- Esther L. Meerwijk conceived and designed the experiments, performed the experiments, analyzed the data, contributed reagents/materials/analysis tools, wrote the paper, prepared figures and/or tables, reviewed drafts of the paper.
- Sandra J. Weiss conceived and designed the experiments, wrote the paper, reviewed drafts of the paper.

### Human Ethics

The following information was supplied relating to ethical approvals (i.e., approving body and any reference numbers):

The Institutional Review Board at the University of California, San Francisco, approved the study.

### Data Availability

The raw data has been supplied as a Supplemental File, and is also available at https://www.researchgate.net/publication/315451654_US_general_population_adults_psychological_pain_survey (Researchgate: DOI: 10.13140/RG.2.2.27299.25127).

### Supplemental Information

Supplemental information for this article can be found online at http://dx.doi.org/10.7717/peerj.3491#supplemental-information.

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
