# Peer review of "Utility of a time frame in assessing psychological pain and suicide ideation"

_PeerJ, doi:10.7717/peerj.3491_

## Round 0.1 · original submission · Major Revisions

Reviewers 1 and 2 were quite positive about the manuscript, but Reviewer 3 wants some major changes, which I believe are achievable.

Reviewer 1 ·

Basic reporting

No comment.

Experimental design

No comment.

Validity of the findings

No comment.

Additional comments

The authors have presented a study examining the usefulness of adding a time frame to the Psychache scale. The manuscript is clear and well written and the methodology is sound. The analyses are appropriate and the authors are careful to state that the results should be interpreted with caution, given the apparent higher vulnerability of the sample. This reviewer has no further comments to add.

Reviewer 2 ·

Basic reporting

It was a pleasure to read this well-written manuscript. The manuscript meets the criteria of basic reporting.

Experimental design

The authors have clearly specified the aims and hypotheses, described the methodologies with sufficient details, and concisely presented the results. The authors have also discussed the findings and limitations of the study, in consideration of potential bias or imprecision.

However, there are some minor revisions which should be addressed before its publication:

1. Title: the authors may consider using the following title ‘Utility of a time frame on the Psychache Scale in assessing psychological pain and suicidal ideation’ as it may reach a broader audience.
2. Abstract: Under methods section, the authors may remove the second sentence ‘Respondents were randomized to complete the original scale (n=133) or a modified scale with time frame (n=109)’ as it seems to repeat the last sentence in the same paragraph.
3. Lines 39-40, please specify a nonclinical convenience sample. Were they general adults or undergraduate students?
4. Line 160 - subsection Data Analysis, could the authors describe statistical analysis they performed to create table 1 (e.g., number, t-test, chi square test, Fisher’s exact test for expected counts less than 5), 2 (mean and SD), and 3 (how they determined the PS and PS-TF’s sensitivity, specificity, and positive and negative predictive value)? Could the authors include percentage in Table 1?
5. Could the authors specify how they addressed missing data? On Table 2, the number of participants responded to PS-TF does not add up to 109.
6. Could the authors attach Psychache scale in the Appendix?
7. Please specify this study used a between-subject design.
8. On Figure 1, the range on the y-axis could be changed to from 10 to 70.

Validity of the findings

The findings of the study provide initial evidence that including a time frame on the Psychache Scale can provide a more accurate assessment of psychological pain and suicidal ideation. The conclusion are well written.

·

Basic reporting

- Sentences 51 to 56 are really difficult to follow.

- Some of the terms are not clearly defined in the Introduction. E.g., when the authors say ‘suicidal individuals’ (line 35) do they mean people with ideation, intention or who have made an attempt? Also, what do the authors mean when they say ‘suicide risk’ (line 40)?

- In some instances, the authors use the term predict for a historical association (e.g., line 37 to 40), which seems strange. Perhaps the term association or associated would be better?

- It would be helpful if some of the items of the PS could be included in the Methods as a part of the description of the measure. I was unable to find a copy of the scale when doing a simple google search.

Experimental design

The authors should include Confidence Intervals for all of the ROC curve point estimates (e.g., AUC, sensitivity, specificity, PPV, NPV). These will help to provide a sense of the accuracy of the estimates and whether there is, in fact, a difference between the two measures – i.e., if the confidence intervals contain the point estimates of each measure, my understanding is that it's not possible to say thereis a difference. I am not an expert with ROC analyses, but I believe a simple comparison of the point estimates is problematic.

Validity of the findings

The present study is attempting to predict the ‘wish to be dead’ which would seem to be different to ‘suicidal ideation’ (i.e., thoughts about ending your own life), which is different to ‘suicidal intent’ (i.e., motivation to end your own life) and ‘suicidal attempts’ (i.e., an attempt to end your own life). We know from a lot of research that it is very hard to predict suicide attempts (e.g., see Large et al., 2011, Systematic review and meta-analysis of the clinical factors associated with the suicide of psychiatric in-patients; Ryan et al, 2010, clinical decisions in psychiatry should not be based on risk assessment), which isn’t mentioned in the manuscript, and the authors don’t seem to make a strong case for why it might be important to be able to predict the ‘wish to be dead’ with a multi-item questionnaire, when you could get this information via asking the question directly? I may have misunderstood, but I think these issues really need to be acknowledged and discussed in the manuscript.

Additional comments

Thank you for the opportunity to review this manuscript, which reports the results of a study comparing the original Psychache Scale that does not contain a timeframe with a modified version that includes a 1 week timeframe. The study (n = 242) involved recruiting participants via the internet and randomising them to receive either the original version or the modified version. The study also involved administering a range of other demographic questionnaires and gathering data about participants’ current ‘wish to be dead’ (i.e., yes or no) and participants’ history of suicidal ideation and attempts. Analyses revealed no differences between the scales in terms of mean scores when participants experienced psychological pain more than a week ago or in the past week. Some evidence was found for the increased PPV among the revised scale versus the original scale when trying to predict the ‘wish to be dead’. This manuscript is interesting, although I have a couple of concerns with it in its current form which I raise via my specific comments.

---

## Round 0.2 · accepted · Accept

I can confirm that you have carried out all the changes suggested by the reviewers.